# Sustainability of Food Placement in Retailing during the COVID-19 Pandemic

Jelena Končar, Radenko Marić *, Goran Vukmirović and Sonja Vučenović

Faculty of Economics, The University of Novi Sad, 24000 Subotica, Serbia; jelena.koncar@ef.uns.ac.rs (J.K.); goran.vukmirovic@ef.uns.ac.rs (G.V.); sonja.vucenovic@ef.uns.ac.rs (S.V.)
* Correspondence: radenko.maric@ef.uns.ac.rs; Tel.: +381-62-278-559

**Abstract:** This work aims to define the impact of different indicators on the sustainability of food placement in the retail sector, during periods of crisis and emergencies, such as the COVID-19 pandemic. An empirical survey conducted in the Western Balkans (WB) region showed that indicators such as developed infrastructure, consistency, and transparency of the supply chain, skilled workers, costs, food safety, food prices, energy consumption, and changes in consumer needs are statistically significant since they affect the sustainability of food placement in the retail sector. As food placement and the retail sector itself are inseparable from other participants in the food supply chain (FSC), an analysis was conducted at the level of all FSC sectors. The results showed some deviations viewed individually in the sectors of production, physical distribution, wholesale, and retail, and in selected Western Balkan countries. Based on the results obtained, the sustainability model of food placement in the retail sector has been defined. The model will serve as the basis for defining the set of measures and incentives that competent institutions and FSC management need to undertake, to minimize the impact of indicators that endanger sustainability. The originality of the study lies in the fact that it fills the research gap that exists in this subject matter in academic research and studies in the WB region. In addition, some indicators important for food placement have been precisely isolated, with the definition of the intensity of their impact, observed overall at the level of the entire FSC as well as by individual sectors. Guidelines and suggestions for future research are listed in the paper.

**Keywords:** food supply chain (FSC); food placement; retail sector; COVID-19 pandemic; sustainability





## 1. Introduction

The key question of this study is how to achieve the sustainability of food placement, as well as the efficiency and transparency of the overall FSC during the crisis caused by the SARS-Cov2 virus [1,2]. The reason for this question is that the COVID-19 pandemic has significantly impaired the sustainability of the supply chain and the efficiency of product placements in the retail sector, primarily products essential for the quarantine period, such as food, hygiene supplies, medical equipment, and medicines [3]. Panic accumulation and stockpiling of long-lasting food products (flour, oil, sugar, salt, bottled water) and easily perishable foods (meat and meat products, milk and milk products, fish, etc.) caused a delay in the supply chain and shortages in retail stores [4]. The lockdowns that marked the beginning and end of 2020 revealed a change in consumer behavior patterns which resulted in a major disparity between supply and demand, delays and errors in deliveries, impaired conditions of product safety, and difficulty in storing and keeping supplies, especially in countries and regions with underdeveloped infrastructure and a lack of transparency in the supply chain [5].

This issue has been addressed by several studies that have found that selected global anti-pandemic measures (lockdowns, physical distance, and work from home) have resulted in major shocks on the demand side, and have burdened the supply chain, primarily food and commodity goods. The global market has seen a significant increase in household food costs by about 30% [6], the accumulation of food products [7], and panic purchases [8].

The mass closure of restaurants, fast food restaurants, bakeries, etc., has led to a sharp decline in food placements towards these segments of the market and the need to reorient the supply chain towards the larger distribution of food to grocery stores, which requires repackaging and processing of products [9]. Along the whole supply chain, costs increased due to the implementation of stricter security measures and regulations, such as mandatory physical barriers in stores, new labeling rules, labeling and packaging of products, contactless distribution, and high consumption of disinfectants [9]. An additional problem that slows down food placement is an increasingly stricter hygiene practice that involves educating employees about food and packaging handling procedures, efficient and frequent cleaning of "high touch" areas (taps, doorknobs, refrigerator handles, kitchen, and restaurant work areas), and thorough washing of fruits and vegetables [10].

A growing challenge is a preorientation of business activities and food delivery to vulnerable groups such as the elderly, the sick, and people in self-isolation. Additionally, the pandemic has changed consumer behavior patterns during purchase, as well as their satisfaction after purchase. Studies [1] show that satisfaction is rapidly declining with the introduction of stricter measures such as mandatory physical distance and wearing protective equipment (such as masks and gloves).

Over time, developed markets dominated by retail chains and a multichannel approach in product placement have managed to neutralize large fluctuations and make the necessary amounts of food available. On the other hand, rural regions with SMEs (Small and Medium-sized Enterprises) as the dominant segment of supply are recording shortages and the untimely supply of food and other essential products. The COVID-19 pandemic will reshape food supply chains significantly, with a special emphasis on faster adjustments to market shocks. The necessity of a multichannel approach in the supply of markets will be more and more emphasized. A study conducted in Germany [10] shows that most German retailers have switched exclusively to online ordering and food delivery. This type of reorientation is due to the redistribution of workers, and additional qualifications.

This work focuses on a sustainability analysis of the food supply chain (FSC) during the COVID-19 pandemic. The authors identify the most important sustainability problems which arose in the FSC as infrastructure [11], consistency and transparency of the supply chain [5], a qualified workforce [12], costs and finances [13], food safety and security [4], food price [14], energy consumption, and changes in consumer needs [1]. The work aims to define the impact of the infrastructure, consistency, and transparency of the supply chain, workforce, costs, food safety and security, food price, energy consumption, and changes in consumer needs on the sustainability of the FSC in the retail sector during the COVID-19 pandemic. For this purpose, an empirical survey was conducted on a sample of 233 FSC managers in the Western Balkans region.

The originality of the study is in its focus on the Western Balkans region. It primarily aims to fill the research gap due to a lack of academic studies dealing with the problem. Only a few papers [15–17] analyze the sustainability and economic consequences of the food placement crisis, issues related to food quality and safety, and the ongoing conflict between supply and demand. Some indicators important for food placement have been precisely isolated, with the definition of the intensity of their impact, observed overall at the level of the entire FSC, as well as by sectors.

In addition to academic interest, the six-state Western Balkans region (Serbia, Croatia, Bosnia and Herzegovina, Montenegro, Northern Macedonia, and Albania) is fit for research because these regions are characterized by different levels of economic and cultural development. This ensures the objectivity of the research approach and the representativeness of the sample. The retail sector comprises the priority economic activity of the Western Balkans, with an 11% share of the total GDP, while the number of registered food retailers is between 15% and 20% of all registered legal entities. Since the production and placement of food products is one of the main drivers of regional economic development, there is a need for a thorough analysis of the economic consequences of the COVID-19 pandemic in this sector. At the same time, support for FSC sustainability is becoming an increasingly

regional issue through numerous financial (tax relief, lending, and financial assistance) and organizational (standardization, monitoring measures, procedures, modern IT solutions, and logistical support) incentives undertaken by WB countries.

Based on the results obtained, the model of sustainability of food placement in the retail sector is defined in the conditions of the crisis. Based on the model, we will define a set of measures and incentives, to support the sustainability of the supply chain, that reflects the practical contribution of the work because it provides guidance that the competent institutions and management of the FSC should undertake to minimize the impact of indicators that pose a threat to sustainability.

The paper consists of five parts. Following the introduction, Theoretical Backgrounds and Hypothesis Development highlights the importance of FSC sustainability, defines indicators crucial to sustainability, and explains the research hypothesis. The research goal, sample, selected statistical method, research procedure, and data processing are an integral part of the Methodology section. The results of the research and testing of the hypothesis are presented in the section entitled Research Results. Discussion contains interpretations of the results obtained, a comparison with the results of similar studies, and defines measures and incentives to support sustainability. Conclusion summarizes the most important results of the survey, conclusions, the shortcomings of conducted research and analysis, and the proposals and guidelines for future research.

## 2. Theoretical Backgrounds and Hypothesis Development

The consequences of the COVID-19 pandemic on the economy, producers, market, and consumers are far-reaching. Lockdown, restriction of movement, employees' illness, and self-isolation directly affect the placement policy, primarily of household products and food. Studies show that the supply chain is not flexible enough to respond to large fluctuations occurring on the demand side [18]. A just-in-time approach in food distribution, with relatively low stocks and a continuous flow of products as a basis for the efficiency of this sector, is not feasible in circumstances of sudden shocks. The situation is particularly alarming in the food placement sector. Grocery stores have faced excessive purchases as a result of people stockpiling fresh and long-lasting groceries. In countries where the number of newly-infected people continues to grow (UK, France, Germany, USA, Brazil, etc.), especially January–March 2021, food shortages seem to peak every day. In addition, there is a drastic trend of increasing demand for personal hygiene products and household cleaning products, medicines, paper haberdashery, bottled water, etc. At the same time, the data show a significant decline in the turnover of non-essential products [19]. Although in some countries (China, South Korea), availability has been normalized for most categories of consumer products (except for medicines), various studies indicate that consumers still, after the lockdown period (January–April 2020), feel concerned about returning to stores [20,21]. The biggest consumer problems are distrust in protection and safety mechanisms in retail facilities. This has caused a drastic drop in sales in traditional grocery stores ranging from −32% (China) to as much as −80% (Mexico).

A particular problem is the distribution of food through global supply chains. Studies show that in a pandemic, localization is of the utmost importance for the sustainability of the chain. Local production can provide a rapid response to local needs with low consumption of resources and energy [22]. On the other hand, the delivery of raw materials from hotspots, retention of products at the borders, food contamination, and infection of employees during transport slows down the placement of goods and response of the supply chain to increased demand. There is also a psychological effect caused by food shortages in the form of panic, fear, and uncertainty among consumers [23].

At the same time, even though it significantly limited traditional retail, the COVID-19 pandemic was a turning point for the expansion of multichannel placement. Physical distance and limited contact have led to a rapid increase in product placement through electronic channels. A total of 52% of consumers avoid brick-and-mortar shopping for fear of infection, while 36% of them do not want to visit traditional stores and go into

crowds until they get the vaccine. Walmart's e-retail of food and consumer goods for households increased by 74% compared to 2019 [24]. Data from the retail chain Mercator showed that in April and May 2020, e-commerce achieved a growth rate of over 40% compared to the months before the pandemic, measured by the total number of completed transactions. The Tehnomanija chain recorded an increase in the number of online users of the Tehnomanija webshop, where 100,000 new users were registered during March alone, with an electronic traffic growth rate of 50% compared to the period before the coronavirus. However, despite these data, consumers state that e-commerce lacks the brick-and-mortar shopping experience and satisfaction with the purchase they get from buying traditionally [1].

Keeping in mind the presented aspects, the indicators that are considered crucial for food placement and consistency of the supply chain in the conditions of market shocks are:

(1) FSC infrastructure [11,25]: represents the basic elements of logistics and transport such as means of transport and transport infrastructure, which ensure the timely functioning of the chain and the whole economic system. Developed infrastructure enables better connectivity between chain participants, more efficient food transport, and timely delivery. The elements of the infrastructure are interconnected and as such must function well at the global level. By analyzing the different approaches of defining and classifying the chain infrastructure, we can conclude that infrastructure includes facilities that ensure food and water supply, agricultural work, health and emergency services, energy availability (electricity, nuclear, gas, and oil), undisturbed traffic (air, road, railway, ports, and waterways), a developed information and telecommunication network, operations of the banking and financial sectors, chemical plants, post offices, etc. [25].

(2) FSC consistency and transparency [5,26]:in recent years, the management of the FSC has undergone significant changes reflecting the increasing globalization of the chain. Food products are traded in large quantities coming from global locations to minimize costs. That complicates supply chain management due to the increasing number of entities involved, which increases the chain's vulnerability and directly affects food placement. There are problems of transparency related to the slow exchange of information among the chain participants, the inability to identify critical points, difficulty monitoring products up and down, issues related to safety and application of quality standards, etc. All of this undermines the sustainability of the FSC, which is already aggravated by pandemic conditions.

(3) Skilled workforce: forms the basis of the functioning of the FSC. Due to the growth of e-retail, Walmart hired 10,000 new workers during 2020, and, for 90,000 existing ones, introduced bonuses on salaries, allowances for overtime and risky work, improved health care, etc. Loblaws provides a salary increase of about 15 percent for store and distribution center workers who continue to work amid the COVID-19 pandemic. However, the data that in 2020, in developing countries, one-third of employees in the sector of food preparation and service (restaurants, bakeries, fast-food restaurants) became unemployed or in the work freeze status, is worrying. Supply chains also suffer losses due to workers' illness, self-isolation of employees, inefficient work from home, etc. Some authors [27] see the solution in the complete digitalization of FSC by implementing IoT.

(4) FSC operating costs [13]: anti-pandemic measures increase costs and slow down food sales. Transaction costs rise due to export restrictions and new export barriers, costs of seeking and contracting food from critical areas, delays at borders, complications in the final stage of delivery, etc. [13]. Furthermore, the volume of investments in FSC is reduced. There is a redirection of funds to other sectors (tourism, health care, purchase of vaccines), and a redirection towards the development of food security systems, etc.

(5) Food safety and security: the pandemic directly affects the four pillars of food safety and security which are availability (whether there is a sufficient supply of food),

access (whether food is available), use (whether food has the necessary nutrients), and stability (whether food is always available) [28]. There are problems such as interruption of food availability, shifting demand to cheaper and less nutritious food, instability of food prices, food contamination, etc. [4]. Food itself is not a way of transmitting the virus [29]. However, potential infection sources occur during its preparation and delivery. Precaution measures in the food sector that directly affect the FSC involve the workers' health (work from home, and self-isolation), personal hygiene (handwashing), frequent disinfection of surfaces and equipment for food preparation, maintaining a clean working environment, mandatory personal protective equipment (masks, and gloves), social distancing, etc. Although these measures are applied in all phases of FSC, the most precautionary measures are crucial in the last phase, namely, consumption [4].

(6) Food price: is closely related to the restriction of consumer movements and changes in their needs. Akter's research [14] shows that countries that opted for stricter measures of lockdown and restriction of movement recorded, only in March 2020, inflation of food prices of about 1%. The most affected are the price indices of meat, vegetables, oil, flour, and fish. The direct impact on the growth of food prices has been growth of transport costs (growth of container transport prices on China–EU routes of over 300%), increase in costs of per diems, accommodation, and quarantine due to slow food flow throughout the FSC, and reduced transport efficiency because fewer quantities are transported in relation to optimal technical capacities.

(7) Energy consumption [30]: studies show a direct correlation between energy consumption and lockdown. Restrictions on movement and reduced international transport (especially air transport and water transport) resulted in a reduction in total gas emissions and fuel consumption. This is supported by the fluctuations in the prices of oil and oil derivatives, which were, at the historical minimum during April 2020, $-0.75$ \$ per barrel as a result of reduced demand in the logistics sector.

(8) A change in consumer needs: is an indicator that most correlates with the consequences of the COVID-19 pandemic. The volume of sales of basic foodstuffs is increasing sharply. The global market of food, medicines, and medical equipment in March 2020 recorded an increase in placements of as much as 100% compared to February. Due to the fear of closing stores, the consumer cart mostly contains products that are essential for consumption, and consumers purchase them in large quantities, for example, meat and meat products, salt, sugar, flour, oil, rice, yeast, dough, canned food, hygiene supplies, pharmaceuticals, medicines, and medical equipment. The share of these products in the total placement is 68.5%. At the same time, the demand for other retail categories decreased significantly (electronic devices $-20.9\%$, clothing $-35.4\%$, footwear $-31.1\%$, furniture $-27.8\%$, and IT equipment and devices $-22.7\%$) [31].

Studies evaluate these impacts differently. For example, Dannenberg et al. [11] and Košanin [25] consider the development of transport infrastructure to be a key indicator. For Mollenkopf et al. [5] and Mattevi [26] it is the transparency of FSC, Wilson [12] sees employees as the fundamental driving force of FSC sustainability, while Cardwell and Ghazalian [13] hold that the key indicator is the cost that occurs in all FSC sectors. Accordingly, the impact of these eight indicators will be tested as individual hypotheses.

**Hypothesis 1 (H1a).** *Infrastructure has a statistically significant impact on the sustainability of food placement in crisis conditions.*

**Hypothesis 1 (H1b).** *The consistency and transparency of FSC has a statistically significant effect on the sustainability of food placement in crisis conditions.*

**Hypothesis 1 (H1c).** *The workforce has a statistically significant effect on the sustainability of food placement in crisis conditions.*

**Hypothesis 1 (H1d).** *The operating costs of the FSC and the level of financial investment have a statistically significant impact on the sustainability of food placement in crisis conditions.*

**Hypothesis 1 (H1e).** *Food safety and security have a statistically significant effect on the sustainability of food placement in crisis conditions.*

**Hypothesis 1 (H1f).** *The price of food has a statistically significant effect on the sustainability of food placement in crisis conditions.*

**Hypothesis 1 (H1g).** *Energy consumption has a statistically significant effect on the sustainability of food placement in crisis conditions.*

**Hypothesis 1 (H1h).** *The changing needs and purchasing habits of consumers have a statistically significant impact on the sustainability of food placement in crisis conditions.*

The literature reveals the different impacts made by these indicators on the sustainability of food placement depend on FSC participants (sectors). Given that four basic sectors (production, physical distribution, wholesale, and retail) participate in food placement, some studies emphasize the different impact of these indicators on the sustainability of each sector [18,22,27]. It is necessary to test whether the differences between FSC participants affect the differences in the impact of these indicators.

The second research hypothesis:

**Hypothesis 2 (H2).** *The differences between FSC participants are statistically significant in predicting differences which appear in the impact of infrastructure, FSC transparency, workforce, costs, food safety, prices, energy consumption, and consumer needs on the sustainability of food placement.*

Some studies show that different degrees of concentration and development of the retail market [32,33], as well as different approaches to sustainability issues, and legislative regulation of this area [27], affect the efficiency of the FSC. As the WB region consists of six countries of different economic and social development, it is necessary to test the following hypothesis regarding the correlation between a specific market segment and the sustainability of the FSC.

The third research hypothesis:

**Hypothesis 3 (H3).** *The differences between the analyzed WB countries are statistically significant in predicting the differences that occur in the impact of infrastructure, FSC transparency, workforce, costs, food safety, prices, energy consumption, and consumer needs on the sustainability of food placement.*

The aim of testing this hypothesis is to define the importance of different market conditions for sustainability, and especially the importance of choosing more or less restrictive anti-pandemic measures in response to each of the analyzed countries to curb the pandemic.

The scientific contribution of this paper is reflected in the fact that the research combines all indicators and compares their individual impact in a crisis situation. The goal of this paper is an empirical study that specifies the significance of each indicator during the COVID-19 pandemic. Accordingly, the impact of these eight indicators will be tested as a separate hypothesis. The analysis was performed separately by FSC participants (sectors) and WB countries (Figure 1).

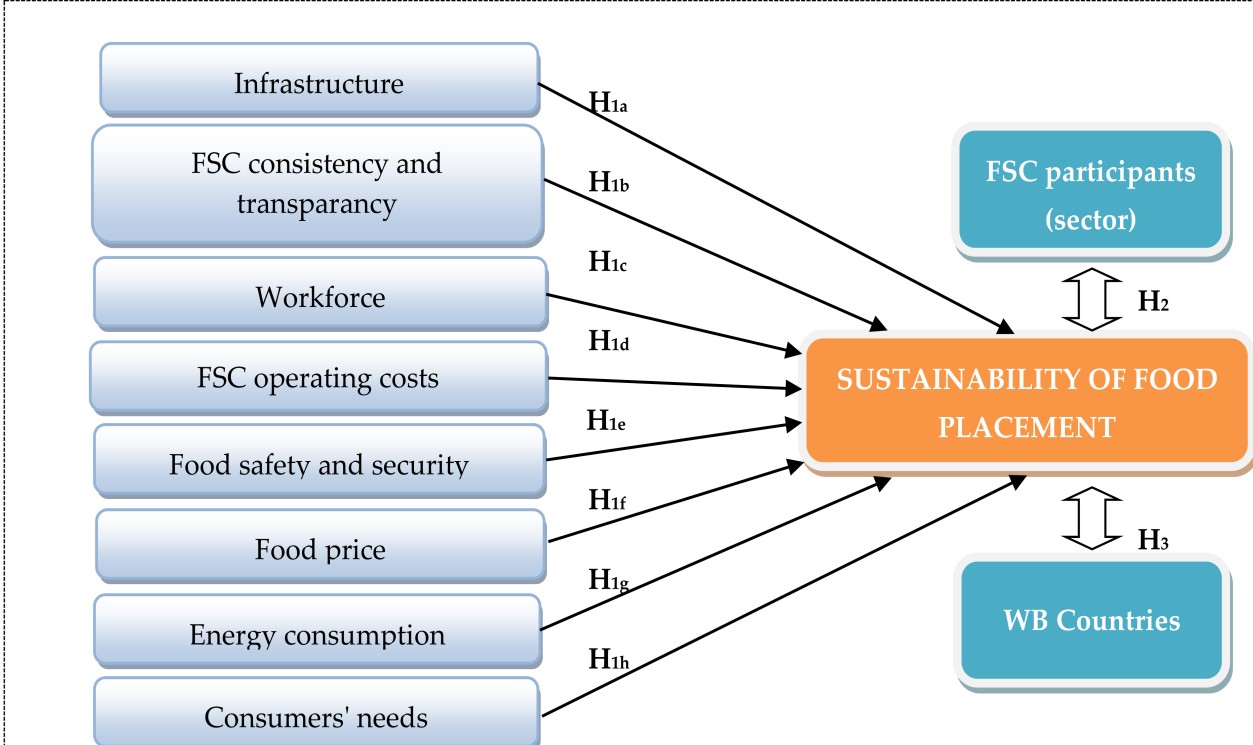

**Figure 1.** FSC consistency and transparency.

## 3. Methodology

### 3.1. Aim

Analyzed theoretical views and presented research results of various studies [1,4,5,11,13,14,25–27,30,31] on the challenges of food placement in the COVID-19 pandemic, provide a basis for identifying the most significant indicators for FSC sustainability. The purpose of the research is to determine the degree of existence of a statistically significant correlation between food placement and identified sustainability indicators. In this regard, the main goal of the research is to clearly define the impact of infrastructure indicators, FSC transparency, workforce, costs, food safety, prices, energy consumption, and consumer needs on the sustainability of food placement in the COVID-19 pandemic.

### 3.2. Research Variables

In accordance with the set hypotheses, the research includes one dependent and several independent variables. The dependent variable is the sustainability of food placement in the retail sector, which is assessed based on the impact of independent variables. Independent variables appear in the form of interval-type measurements and as grouping variables. Independent variables of the interval type of measurement are the following indicators: infrastructure, FSC consistency and transparency, workforce, costs, food safety and security, price, energy consumption, and consumer needs [1,4,5,11,13,14,25–27,30,31]. The independent grouping variables are the selected WB countries (Serbia, Croatia, Bosnia and Herzegovina, Northern Macedonia, and Montenegro) and the participants (sectors) of the FSC (production, physical distribution, wholesale, and retail).

### 3.3. Research Sample

The sample consisted of 233 FSC managers from selected WB countries. The representativeness of the respondents is reflected in the fact that the managers directly responsible for food placement, the managers of retail facilities, and large distribution systems for storage and storage of food products, were surveyed. This methodology collected data about real challenges to the sustainability of food placement in the conditions of the COVID-19

pandemic. In addition, the selected number of FSC managers falls into the category of a large statistical sample. Since the respondents come from different sectors of the FSC, and different WB countries, their number is uniform according to the analyzed strata. Table 1 illustrates the detailed structure of the research sample.

**Table 1.** Research sample structure.

| FSC Sector | *n* | Structure (%) |
|---|---|---|
| Production | 67 | 28.8% |
| Physical Distribution | 52 | 22.3% |
| Wholesale | 59 | 25.3% |
| Retail | 55 | 23.6% |
| **WB Countries** | ***n*** | **Structure (%)** |
| Croatia | 51 | 21.9% |
| Serbia | 55 | 23.6% |
| Montenegro | 41 | 17.6% |
| Bosnia and Hercegovina | 42 | 18.0% |
| North Macedonia | 46 | 19.7% |

Source: the authors' calculation.

### 3.4. Research Procedure and Data Analysis

The study was conducted within nine months of the first case of COVID-19 in the WB region (6 March 2020). The period between March and December 2020 was marked by the declaration of the state of emergency and lockdown policies in all countries in the region. In line with anti-pandemic measures, the research was conducted based on an e-questionnaire that was forwarded to FSC participants. The questionnaire was created based on the questions defined in the questionnaires used in previous studies [1,4,5,11,13,14,25–27,30,31]. After the general information about the respondents (FSC sector, WB countries), the respondents, using the Likert-type measurement scale (0-no significance; 5-very high significance), ranked the impact of each indicator on the sustainability of food placement in the COVID-19 pandemic. Each indicator was tested based on three items.

Infrastructure is defined as the ability to respond to demands during a pandemic, the degree of information and telecommunications network developed, and good connectivity between FSC participants.

FSC consistency and transparency contains respondents' views on satisfaction with the information exchange, and the ability to identify critical points and monitor product flow.

The workforce includes the respondents' attitudes regarding the efficiency of work from home, satisfaction with employees' skillset, and the consequences of infection, self-isolation, and social distance.

Operating costs are defined as a reduction in inventory costs, an increase in transaction costs, and a decrease in investment.

Food safety and security was measured by the total amount of food, the amount of food available, and the implementation of precautionary measures in the food sector that directly affect food placement.

The price of food is measured through high costs of per diems, accommodation and quarantine, reduced transport efficiency, and rising costs of physical distribution.

Energy consumption is defined as oil consumption, gas emissions, and fuel consumption due to limited movement and transport.

Consumer needs include respondents' attitudes towards avoiding traditional stores, the growth rate of e-commerce, and the decline in demand for long-lasting products (furniture, footwear, and clothing). The dependent variable (sustainability of food placement in the retail sector) was also tested based on three items.

Sustainability of food placement is defined as the timely availability of food, satisfaction with the existing way of inventory management, and the degree of digitalization of the FSC.

Managers from the largest FSC systems (Lidl, Delta Transport Logistics, Imlek, Banat Oil Factory, Univerexport, Victoriaoil, Dijamant, Mercator, etc.) and small farms in the WB region were surveyed. The questionnaire was sent to the addresses of 277 FSC managers. The response rate was 84%.

Table 2 presents the reliability and correctness of the selected scales using the internal consistency Cronbach's alpha coefficient, the Skewness coefficient showing the symmetry of the distribution, and the Kurtosis coefficient as measures of the distribution of results. Given that the values of Cronbach's alpha for all indicators were above 0.700 and that no statistically significant deviations were read when applying the Skewness and Kurtosis coefficients, the reliability and correctness of the selected scales were confirmed. That is, it was confirmed that the selected questions describe an identical problem and can be used to examine the views of FSC managers.

**Table 2.** Cronbach's Alpha, Skewness, and Kurtosis coefficient.

| Indicators | Cronbach's Alpha | Skewness | Kurtosis |
|---|---|---|---|
| Infrastructure | 0.822 | 0.058 | −1.031 |
| FSC consistency and transparency | 0.839 | −0.119 | −1.028 |
| Workforce | 0.728 | 0.331 | −0.994 |
| FSC operating costs | 0.802 | −0.438 | −1.162 |
| Food safety and security | 0.752 | −0.121 | −1.327 |
| Food price | 0.955 | 0.068 | −1.418 |
| Energy consumption | 0.923 | −0.229 | −0.636 |
| Consumers' needs | 0.705 | 0.384 | −0.443 |

Source: the authors' calculation.

The SPSS 20 statistical package was used to process the collected data. The descriptive statistics method was used to describe the statistical indicators of the sample. Structural modeling (SEM) or path analysis was used to test the hypothesis on the impact of individual variables on the sustainability of food placement, while the multiple regression analysis method was used to test the correlation between the FSC sector and the WB countries.

## 4. Research Results

The average ranking of respondents' agreement with the statement that infrastructure, FSC transparency, workforce, costs, food safety, price, energy consumption, and consumer needs affect the sustainability of food placement in the COVID-19 pandemic is presented in Table 3. The average rankings for each of the tested indicators, as well as the most important descriptive indicators, are displayed in the table.

**Table 3.** Descriptive statistics for selected indicators.

| Order No. | Indicators | Min. | Max. | Mean (M) | Standard Error (SE) | Standard Deviation (SD) |
|---|---|---|---|---|---|---|
| 1 | 2 | 3 | 4 | 5 | 6 | 7 |
| 1 | Infrastructure | 2.00 | 5.00 | 3.82 | 0.0714 | 1.1715 |
| 2 | FSC consistency and transparency | 1.00 | 5.00 | 3.66 | 0.0738 | 0.9490 |
| 3 | Workforce | 2.00 | 4.00 | 4.31 | 0.0694 | 0.7124 |
| 4 | FSC operating costs | 1.00 | 5.00 | 4.18 | 0.0583 | 0.8227 |
| 5 | Food safety and security | 2.00 | 5.00 | 4.63 | 0.0900 | 0.8425 |
| 6 | Food price | 1.00 | 5.00 | 4.28 | 0.0740 | 1.2419 |
| 7 | Energy consumption | 2.00 | 5.00 | 3.76 | 0.0557 | 1.0648 |
| 8 | Consumers' needs | 1.00 | 5.00 | 3.90 | 0.1278 | 0.9401 |

Source: the authors' calculation.

The surveyed FSC managers responded that the greatest importance for the sustainability of food placement is the safety and security of food products (M = 4.63). This result is expected, given the fact that the COVID-19 pandemic, on the one hand, has caused much stricter measures and safety standards when moving food from ascending to descending participants in the supply chain. On the other hand, there is a risk of the virus directly contaminating food. All of this complicates the placement and leads to stagnation and the creation of critical points in the FSC. Respondents mostly agreed (SD = 0.7124) that the sustainability of placements and the functioning of the entire FSC depends on employees. The workforce indicator ranked second in importance (M = 4.31). Cases of infection, self-isolation of employees, and quarantine measures are reflected in the efficiency of the functioning of the FSC and impose an increasing need for digitalization of business activities of the chain. Respondents cited the price of food (M = 4.28) and the FSC operational costs (M = 4.18) as significant indicators. The price of food products essential for the lockdown period (long-term canned food, flour, oil, yeast, rice, etc.), has risen sharply due to market shock and a demand growth for these products of over 100% [14]. Due to certain export barriers, food procurement from areas marked as the epicenter of the epidemic, safety protocols, and stockpiling at final points of sale, FSC operational costs are rising, which is also reflected in the slowdown in food sales [13]. With this view, the respondents showed an extremely high degree of agreement (SD = 0.8227). Consumer needs (M = 3.90) undoubtedly complicate the functioning of the FSC. Panic and the accumulation of food, medicine, protective and medical equipment leads to supply disruptions, large gaps in demand, and shortages. FSC managers attached somewhat less importance to energy consumption, infrastructure, and transparency of the FSC.

To gain a more precise insight into the importance of these indicators on the sustainability of food placement, the first group of research hypotheses **H1a–H1h** was tested using the structural modeling (SEM) method. Before applying the SEM method, multiple regression analysis first determined the degree of correlation between the observed indicators and sustainability. The enter method was applied combining all independent variables into the prediction of the dependent variable (sustainability of food placement in the retail sector). The obtained regression model is statistically significant ($F_{(200, 7)}$ = 2.639, $p < 0.01$), which implies that the set of examined indicators statistically significantly predicts the sustainability of food placement in the WB region. The model set up in this way describes 71.2% of the variance of the criteria. Table 4 presents the contribution of each indicator (predictor).

**Table 4.** Testing the contribution of individual indicators.

| | Stand. Coefficient | | t | Sig. |
|---|---|---|---|---|
| | **Beta** | **St. Error** | | |
| (const.) | 0.679 | 1.072 | 3.457 | 0.000 |
| Infrastructure | −0.043 | 0.471 | 0.639 | 0.067 |
| FSC consistency and transparency | −0.521 | 0.584 | 0.712 | 0.149 |
| Workforce | 0.762 ** | 0.503 | 1.020 | 0.001 |
| FSC operating costs | 0.518 * | 0.611 | 1.127 | 0.044 |
| Food safety and security | 0.829 ** | 0.870 | 0.386 | 0.017 |
| Food price | 0.643 ** | 0.567 | 1.240 | 0.007 |
| Energy consumption | −0.276 | 0.694 | 0.504 | 0.284 |
| Consumers' needs | 0.443 * | 0.944 | 0.681 | 0.034 |

Note: * correlation is significant at the level 5%, ** correlation is significant at the level 1%. Source: the authors' calculation.

The results show that food safety and security (B = 0.829; $p < 0.01$), workforce (B = 0.762; $p < 0.01$), food price (B = 0.643; $p < 0.01$), FSC operating costs (B = 0.518; $p < 0.05$), and consumers' needs (B = 0.443; $p < 0.05$) statistically significantly predict the sustainability of food placement during the COVID-19 pandemic in the WB region. The results coincide with the previous studies [1,4,5,11,13,14], which consider the above indica-

tors to be the most responsible for the sustainability of the FSC. The correlation between the tested indicators and the sustainability of food placement is positive, which means that with the increase in the intensity of the indicators, the sustainability of placement and the functioning of the entire FSC is impaired. Transparency of FSC ($p > 0.05$), energy consumption ($p > 0.05$), and infrastructure ($p > 0.05$) show slightly less importance for predicting sustainability.

As the focus of the research is on defining the impact that indicators have on the sustainability of food placement, the SEM method was applied, and we can see based on defined pathways how these indicators affect sustainability. The results of the path analysis show that the fitting of the model is satisfactory (NFI = 0.966, RFI = 0.928, IFI = 0.988, TLI = 0.975, CFI = 0.988, RMSEA = 0.039, and CMIN/DF = 1.497). Table 5 presents a model that defines the impact of given indicators on the sustainability of food placement, and a statistically significant interaction between the indicators themselves.

**Table 5.** Results of path analysis.

| Ord. No. | Path | Path Coefficient | T Value | Result |
|---|---|---|---|---|
| 1 | 2 | 5 | 6 | 7 |
| 1 | Workforce » Sustainability of food placement | 0.771 | 15.221 | Support |
| 2 | Food safety and security » Sustainability of food placement | 0.166 | 3.077 | Support |
| 3 | Consumers' needs » Food safety and security | 0.539 | 10.03 | Support |
| 4 | Consumers' needs » Sustainability of food placement | 0.099 | 1.835 | Reject |
| 5 | Consumers' needs » Food price | 0.706 | 11.250 | Support |
| 6 | FSC operating costs » Sustainability of food placement | 0.071 | 1.718 | Reject |
| 7 | FSC operating costs » Food price | 0.277 | 3.144 | Support |
| 8 | Food price » Sustainability of food placement | 0.648 | 9.033 | Support |
| 9 | Infrastructure » FSC operating costs | 0.184 | 3.527 | Support |
| 10 | Energy consumption » FSC operating costs | 0.537 | 8.303 | Support |
| 11 | Infrastructure » Sustainability of food placement | 0.054 | 0.998 | Reject |
| 12 | Energy consumption » Sustainability of food placement | 0.032 | 1.200 | Reject |
| 13 | FSC consistency and transparency FSC » Sustainability of food placement | 0.012 | 1.609 | Reject |

Source: the authors' calculation.

We can confirm, based on the presented model, that the sustainability of food placement is under the direct influence of the workforce, procedures related to food safety and security, and food prices. This confirmed the research hypotheses **H1c**, **H1e**, and **H1f**. Consumer needs have a statistically significant impact on food safety and security and food prices. Thus, the needs of consumers, indirectly, through the mentioned indicators, affect the sustainability of placements. There is also a statistically significant impact of FSC operating costs on food prices, and a direct impact of infrastructure and energy consumption on FSC costs. No statistically significant effect was detected for other indicators, so the research hypotheses **H1a**, **H1b**, **H1d**, **H1e**, and **H1h** were rejected. Figure 2 illustrates the structural model or ways of influencing the analyzed indicators on the sustainability of food placement.

As the FSC consists of several different sectors, it is necessary to test whether the differences that occur in the impact of these indicators on the sustainability of food placement are statistically significantly related to differences between FSC participants, i.e., between the sectors of production, physical distribution, wholesale, and retail. To test the second research hypothesis **H2**, which examines this correlation, the multiple regression analysis was applied. The enter method was applied, which for each FSC sector combines all independent variables into the prediction of the dependent variable (sustainability of food placement in the retail sector). The obtained regression model is statistically significant for each of the observed participants. For production ($F_{(120, 7)}$ = 2.792, $p < 0.01$), the model describes 62.4% of the variance of the criteria. For physical distribution ($F_{(60, 7)}$ = 2.953, $p < 0.01$), the model describes 57.2% of the variance of the criteria.

For wholesale ($F_{(60, 7)} = 2.166$, $p < 0.05$), the model describes 41.2% of the variance of the criteria, and for retail ($F_{(60, 7)} = 2.953$, $p < 0.01$), the model describes 73.1% of the variance of the criteria. Table 6 presents the contribution of individual indicators (predictors) by sectors.

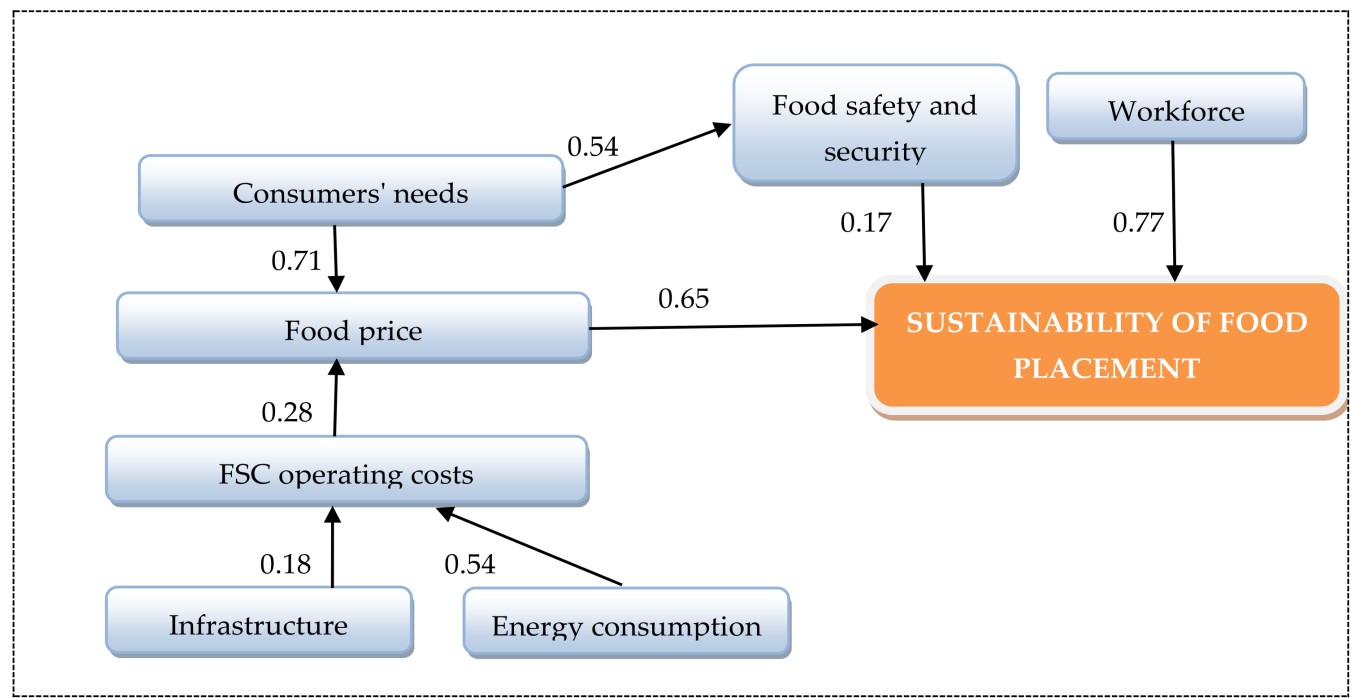

**Figure 2.** Structural model.

**Table 6.** Contribution of individual indicators by FSC sectors.

| | Production | | Physical Distribution | | Wholesale | | Retail | |
|---|---|---|---|---|---|---|---|---|
| | Beta | Sig. | Beta | Sig. | Beta | Sig. | Beta | Sig. |
| (const.) | 0.72 | 0.039 | 0.83 | 0.000 | 0.52 | 0.040 | 0.44 | 0.001 |
| Infrastructure | 0.652 | 0.081 | 0.512 ** | 0.003 | 0.771 * | 0.033 | 0.684 | 0.122 |
| FSC consistency and transparency | −0.597 | 0.275 | 0.818 | 0.044 | 0.828 | −0.081 | 0.512 | 0.668 |
| Workforce | 0.580 ** | 0.007 | 0.738 ** | 0.000 | 0.813 ** | 0.000 | 0.857 * | 0.031 |
| FSC operating costs | 0.334 * | 0.034 | 0.475 | 0.870 | 0.447 ** | 0.003 | 0.085 | 0.143 |
| Food safety and security | 0.582 ** | 0.000 | 0.602 ** | 0.003 | 0.764 | 0.694 | 0.803 ** | 0.000 |
| Food price | 0.838 ** | 0.008 | 0.815 * | 0.007 | 0.935 ** | 0.002 | 0.839 ** | 0.001 |
| Energy consumption | 0.611 * | 0.032 | 0.668 ** | 0.000 | 0.505 | 0.087 | 0.722 | 0.083 |
| Consumers' needs | −0.780 | 0.421 | 0.671 | −0.336 | 0.416 | 0.066 | 0.529 ** | 0.004 |

Note: * correlation is significant at the level 5%; ** correlation is significant at the level 1%. Source: the authors' calculation.

Observed individually by FSC sectors, workforce, food safety, security, and food prices statistically significantly predict sustainability in each of the sectors. Additional statistical significance appears sporadically, depending on the type of activity. In production, it is energy consumption ($\beta = 0.611$, $p < 0.05$) and operating costs of the FSC ($\beta = 0.334$, $p < 0.05$). In the physical distribution sector, it is infrastructure ($\beta = 0.512$, $p < 0.01$) and energy consumption ($\beta = 0.668$, $p < 0.01$). Wholesalers considered infrastructure to be a significant indicator that predicts sustainability ($\beta = 0.771$, $p < 0.05$), while for retailers this was a sharp change in consumer needs ($\beta = 0.529$, $p < 0.01$). The remaining indicators do not show statistical significance. Other analyzed indicators do not have a statistically significant impact. The conducted testing confirmed hypothesis **H2** that the differences between the FSC sectors were statistically significantly related to the differences in the impact of

indicators on the sustainability of food placement. That is, differences in the impacts of indicators on sustainability are different among major FSC participants in the WB region during the COVID-19 pandemic. The finding confirms the results of previous research on the significance of differences between business activities within the FSC [33–38].

The influence of indicators in each of the sectors was defined by SEM. The results of the path analysis show that the fitting of the model for all sectors is satisfactory (p: NFI = 0.994, RFI = 0.918, IFI = 0.977, TLI = 0.961, CFI = 0.973, RMSEA = 0.048, CMIN/DF = 1.386), (phd: NFI = 0.922, RFI = 0.927, IFI = 0.964, TLI = 0.928, CFI = 0.913, RMSEA = 0.027, CMIN/DF = 1.479), (w: NFI = 0.977, RFI = 0.951, IFI = 0.988, TLI = 0.957f, CFI = 0.988, RMSEA = 0.034, CMIN/DF = 1.377), and (r: NFI = 0.983, RFI = 0.928, IFI = 0.988, TLI = 0.914, CFI = 0.907, RMSEA = 0.039, CMIN/DF = 1.286). Figure 3 illustrates the structural models or pathways of the impact of indicators on the sustainability of food placement, as observed by FSC sectors.

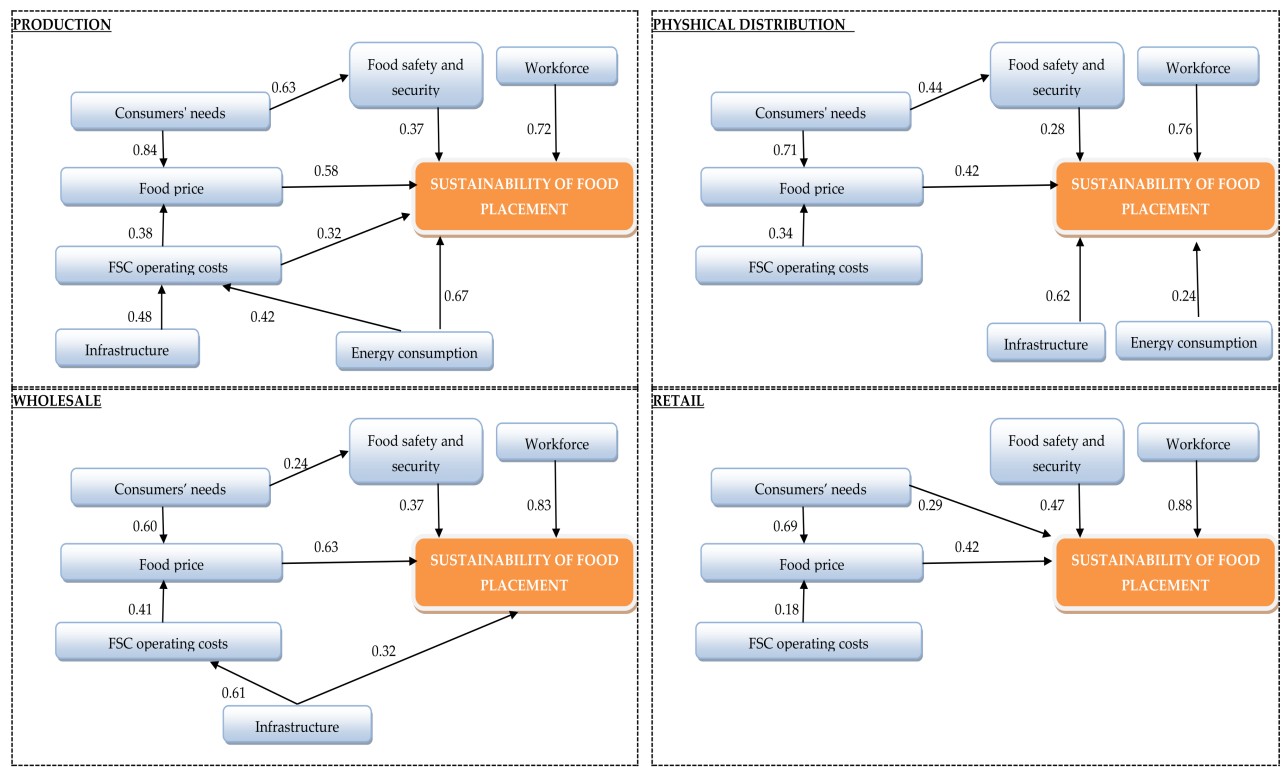

**Figure 3.** Structural models of FSC sectors.

It is evident from the presented models that, depending on the type of business activity of FSC participants, different indicators come to the fore in the conditions of the pandemic. Infrastructure development and better connectivity between FSC participants are a priority in wholesale and physical distribution. Energy consumption is an important item for producers, while large shocks on the demand side and changing patterns of consumer behavior affect retailers the most. They are reflected in shortages, high selling prices, panic purchases, etc.

Finally, it is necessary to test whether there are differences in the impact of indicators on sustainability observed by WB countries. Hypothesis **H3** tests whether differences between countries in the WB region significantly predict differences in the impact of indicators on the sustainability of food placement during the COVID-19 pandemic. The multiple regression analysis was applied. Table 7 presents the test results of individual contributions of the analyzed indicators depending on the WB country.

**Table 7.** Testing the contribution of individual indicators.

| WB Country | Indicators | F | Sign. |
|---|---|---|---|
| Croatia | Infrastructure | 0.758 | 0.188 |
| | FSC consistency and transparency | 1.115 | 0.891 |
| | Workforce | 1.132 | 0.629 |
| | FSC operating costs | 0.730 | 0.079 |
| | Food safety and security | 0.371 | 0.081 |
| | Food price | 0.672 | 0.313 |
| | Energy consumption | 0.127 | 0.662 |
| | Consumers' needs | 0.610 | 0.545 |
| Serbia | Infrastructure | 0.101 | 0.568 |
| | FSC consistency and transparency | 1.714 | 0.047 |
| | Workforce | 0.640 | 0.087 |
| | FSC operating costs | 0.678 | 0.066 |
| | Food safety and security | 1.502 | 0.571 |
| | Food price | 1.066 | 0.077 |
| | Energy consumption | 0.583 | 0.516 |
| | Consumers' needs | 0.818 | 0.456 |
| Montenegro | Infrastructure | 0.820 | 0.268 |
| | FSC consistency and transparency | 1.207 | 0.238 |
| | Workforce | 1.011 | 0.221 |
| | FSC operating costs | 0.922 | 0.210 |
| | Food safety and security | 1.010 | 0.319 |
| | Food price | 0.556 | 0.638 |
| | Energy consumption | 0.876 | 0.079 |
| | Consumers' needs | 0.612 | 0.064 |
| Bosnia and Hercegovina | Infrastructure | 0.215 | 0.817 |
| | FSC consistency and transparency | 1.060 | 0.087 |
| | Workforce | 1.413 | 0.087 |
| | FSC operating costs | 1.337 | 0.442 |
| | Food safety and security | 0.451 | 0.720 |
| | Food price | 2.175 | 0.401 |
| | Energy consumption | 2.451 | 0.055 |
| | Consumers' needs | 1.214 | 0.063 |
| North Macedonia | Infrastructure | 0.536 | 0.066 |
| | FSC consistency and transparency | 1.336 | 0.544 |
| | Workforce | 0.357 | 0.387 |
| | FSC operating costs | 2.518 | 0.144 |
| | Food safety and security | 0.689 | 0.189 |
| | Food price | 0.531 | 0.221 |
| | Energy consumption | 0.457 | 0.511 |
| | Consumers' needs | 0.500 | 0.070 |

Source: the authors' calculation.

From Table 7, it is clear that when observed by selected WB countries, there are no differences in indicators' impact on the sustainability of food placement in the retail sector ($p > 0.05$). The surveyed FSC managers in charge of food placement in different countries of the Western Balkans, identically observed the impact of selected indicators on sustainability issues during the COVID-19 pandemic. This result is partly expected, as all countries in the region have opted for an identical pandemic strategy and opted for standard anti-pandemic measures such as country lockdown, limiting contact, self-isolation, quarantine, and mandatory use of masks and medical equipment. In support of this result is the fact that the study included FSCs that are regional and operate in the WB as a single entity, with a single sustainability policy, and unique procedures and standards of safety, security, and health. Employees of companies such as Lidl, Delta Transport Logistics, Imlek, Banat Oil Factory, Univerexport, Victoriaoil, Dijamant, Mercator, etc., have an identical approach when it comes to implementing anti-pandemic measures, working from home,

self-isolation, food protection from contamination, etc. All of the above is a consequence of the unique procedures at the level of these companies, regardless of the country in which they operate. Keeping in mind the presented aspects, the third research hypothesis **H3** was not accepted.

## 5. Discussion

The results obtained partially confirm some earlier research findings that considered workforce, food prices, and safety and security standards directly responsible for the sustainability of the FSC [4,14,28,29]. The impact of these indicators is especially visible in times of crisis and market shocks caused by the COVID-19 pandemic. The changing patterns of consumer behavior, which raise the price of food due to panic purchases and stockpiling, also have an indirect impact on the sustainability of food sales. At the same time, there are growing health concerns and fear of infection, and safety requirements are tightening procedures and standards for the safe flow of food along the FSC. This confirms the results of the study by Brandtner et al. [1] and Kravela et al. [39,40], who consider consumers to be the most responsible for the concept of sustainability. The operating costs, which have grown exponentially during the COVID-19 pandemic, also have a significant impact on the price of food and thus, indirectly, on the sustainability of the entire chain. International product placement is significantly more expensive, and there are unplanned costs, as well as longer delivery times. Due to various measures imposed by states and regions for the movement of people and goods between countries, there are frequent cases of border congestion, mandatory quarantine, complicated transport procedures, complex documentation, etc. This directly increases the cost of the FSC and, at the same time, the price of food. These results are confirmed by studies by Cardwell and Ghazalian [13] and Akter [14] on the direct correlation between operational costs and food prices.

The conducted research shows a different influence of the given indicators on the food placement among FSC participants. This is also an expected result, given recent research [33–38,41] that highlights the strong correlation of differences in FSC business sectors with placement efficiency. In this context, energy consumption comes to the fore in production, which has no statistically significant impact in retail, but rather the changing needs of consumers. Wholesalers and distributors, on the other hand, consider the level of developed infrastructure crucial.

The obtained results can serve as a guideline for defining a set of measures and incentives that the competent institutions and FSC management should take in order to minimize the impact of indicators that pose a threat to sustainability. We differentiate three groups of anti-pandemic measures to be implemented separately by the analyzed FSC sectors and WB countries.

Financial measures, aimed at economic and fiscal incentives, must be adopted as a form of financial assistance to the most vulnerable FSC participants (small-size shops and independent farmers). These measures should include tax exemptions for certain categories of food products, reduction of taxes on income and property, simplifying the conditions for lending to FSC participants, introducing credit facilities, grants, etc. A specific measure should aim at solving the problem of layoffs. Therefore, we propose financial incentives approval for new employment or maintaining the existing staff capacity. When it comes to producers, financial incentives should focus on the possibility of deferred payment of raw materials, reimbursement of costs incurred due to the retention of raw materials, and intermediate goods at the borders, and credit facilities for the modernization of production processes and plants. When it comes to participants of physical distribution, the key incentive is the reduction of transport costs (abolishing tolls and parking prices), reduction of fuel prices and so on.

Anti-pandemic safety measures refer to further strict implementation of existing measures in the segment of safety and security. These measures should reduce the risk of infecting FSC employees, as well as consumers themselves during sales activities, to a minimum. In the retail sector, this includes stricter compliance with safety and health stan-

dards of employees and consumers, the mandatory wearing of protective masks, regular disinfection of work surfaces and products, regular cleaning, and maximum hygiene of warehouses, shelves, refrigerators, and display cases with perishable products. In addition to all of this, it is necessary to mark the obligatory directions of consumer movement in retail facilities to maintain social distance and prevent contact. In the wholesale sector, the emphasis should be on clear safety protocols that include employee hygiene and mandatory protective equipment when working with perishable food (meat, meat products, milk, dairy products, fruits, vegetables, fish, etc.), cleaning ventilation systems, and regular ventilation of warehouses, use of large barriers and tunnels for disinfection, and constant monitoring of temperature and atmospheric conditions in warehouses. Manufacturers should introduce strict measures to control raw materials and prevent food contamination during packaging and production. Protective equipment is mandatory for all workers who work on the lines and in contact with raw materials and food. It is necessary to conduct regular tests to confirm that the animals used for food production are healthy. When it comes to physical distribution, anti-pandemic safety measures should focus on regular hygiene of the vehicle interior, disinfection and cleaning of loading and unloading machines, avoiding human contact with cargo, etc.

Organizational measures imply the introduction of a new business concept based on the digitalization of all business activities and the shortening of the FSC [42,43]. This set of measures should include: (1) in the manufacturing sector, (a) implementation of modern IT solutions for better flow of information along the FSC such as IoT, biosensors, TT indicators, and RFID technology [44], (b) redirection of employees and business operations to work from home, (c) introduction of clear standards, supervision measures, and procedures for teleworking; (2) in the physical distribution sector, this means the transformation of the traditional FSC to electronic FSC with the expansion of electronic food placement (it should reach a share of 10–15% of total food placement); (3) in the wholesale sector the emphasis is on logistic support for the electronic placement of food in the form of special delivery services; and finally, (4) for the retail sector the promotion of electronic ordering and development of new electronic stores is the key. In addition, the possibility of immediate contact between employees and consumers should be reduced in the retail facilities specialized in food placement [45,46] through non-cash payment, electronic cash registers, etc.

It is necessary to give different priorities to the indicated measures in the analyzed countries because of the imbalanced level of market and economic development across the WB region. As regional economic leaders, Croatia and Serbia need to focus on higher-level compliance to anti-pandemic safety measures. The competent ministries must appoint a body responsible for monitoring the implementation of measures, for example, market inspection in facilities for wholesale and retail. That means implementing a clear system of penalties and responsibilities for violating security protocols and spreading infection. Also, state financial assistance is necessary for the FSC in Montenegro, Bosnia and Herzegovina, and Northern Macedonia. These countries had an enormous drop in GDP in 2020, which also calls into question the sustainability of the FSC itself. Through state aid, in which international institutions should also participate, it is necessary to ensure a minimum of funds for the functioning of the FSC, the existence of employees, and the uninterrupted supply of the market. Furthermore, anti-pandemic safety measures must be constantly monitored and implemented. At the level of the entire WB market, stakeholders should work on continuous transformation from traditional to electronic placement channels through the digitalization of all business processes, which will ensure a supply of domicile markets and prevent a further recession in the WB.

The theoretical contribution of this study lies in the fact that it precisely defines the individual direction of the impact of key indicators of the sustainability of food placement during the crisis period. The indicators' impact is accurately defined depending on the FSC participants and the WB countries analyzed. Consequently, a set of measures and

incentives for the sustainability of food placement was proposed. At the same time, this approach has filled the research gap which existed in academic research in the WB region.

## 6. Conclusions

The need for the conducted research stemmed from the fact that the current pandemic has completely paralyzed certain sectors such as tourism, catering, and retail. The shocks that emerged on the demand side, primarily for food, led to a disruption in the efficiency of the FSC, delays in deliveries, and shortages of essential food products. Hence, it was necessary to examine the impact of certain indicators on the sustainability of placements and the functioning of the entire FSC. The structure of the research sample consisted of respondents employed in the supply chains of the WB region. WB was chosen due to the good institutional cooperation of the authors with leading economic entities, as well as familiarity with regional economic trends.

The conducted research defined the impact of various indicators on the sustainability of food placement. This partially confirmed the first group of research hypotheses, more precisely hypotheses **H1c**, **H1e** and **H1f**. Labor, food prices, and food safety and security are indicators that have a direct impact on the sustainability of food placement during the COVID-19 pandemic. Indicators having an indirect influence on sustainability, such as costs, energy consumption, and infrastructure, have also been defined. It was found that the differences between FSC participants significantly predict the different level and direction of the impact of indicators on sustainability (**H2**). Differences have not been confirmed between WB countries (**H3**), primarily as a result of the implementation of identical anti-pandemic and economic recovery measures in all WB countries. Based on the defined impact of the indicators, a whole set of measures, incentives, and subsidies has been proposed that should be activated in the WB region, to ensure the sustainability of food placement, continuity in delivery, and more efficient functioning of the FSC.

The shortcoming of the research is the geographical limitations of the WB region. The reasons for this choice are the availability of data necessary for analysis and the authors' knowledge of the food placement method in selected countries. Possibly, as a shortcoming, the structure of the questionnaire (survey) could be mentioned, because it consisted of pre-prepared questions, which can lead to simplified conclusions. In addition, there are some other limitations of the research which should be considered: (1) sample limitations to the managerial level—administrative staff, transport workers, line workers, sellers in retail, and staff on packaging lines should all be included in future analyses. They were not included at the time of the research due to safety measures and the inability to conduct such broad research; (2) limitations of the indicators—although this study included the key indicators which impact the sustainability of the food placement, we cannot say with certainty that there are no other indicators to be considered.

Guidelines for future research include expanding the sample of respondents to countries outside the WB and performing a comparative analysis of data obtained in the EU/non-EU countries. In addition to FSC managers, administration and direct executors should be included in the survey (sales staff, warehouse workers, workers on loading and unloading, and drivers). The number of indicators should be expanded to include certain categories of sub-indicators obtained by a broader range of analyses in the available literature in the period of the COVID-19 pandemic. This would complement the scientific view of food placement problems during the COVID-19 pandemic, and better prepare the FSC for the post-COVID period.

**Author Contributions:** Conceptualization, J.K. and G.V.; methodology, R.M.; investigation, R.M. and S.V.; resources, G.V.; data curation, S.V.; writing—original draft preparation, R.M. and S.V.; visualization, R.M.; supervision, J.K. All authors have read and agreed to the published version of the manuscript.

**Funding:** This research received no external funding.

**Institutional Review Board Statement:** Not applicable.

**Informed Consent Statement:** Informed consent was obtained from all subjects involved in the study.

**Data Availability Statement:** The data presented in this study are available in this article.

**Conflicts of Interest:** The authors declare no conflict of interest. The funders had no role in the design of the study; in the collection, analyses, or interpretation of data; in the writing of the manuscript, or in the decision to publish the results.

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
