# Peer review of "Sustainability of Food Placement in Retailing during the COVID-19 Pandemic"

_sustainability, doi:10.3390/su13115956_

Round 1

Reviewer 1 Report

Abstract:
1.    The originality of the study needs to be provided in the abstract section. 

Introduction:
1.    Instead of mentioning some incentives which can be utilized to support the sustainability of the SC (on line 88-89), please explicitly state very important ones. 
2.    Please write the originality of the study in the introduction. 

Theoretical backgrounds and hypothesis development:
1.    Some of the arguments for the first paragraph require date validation. Because due to pandemic, most of the countries has entered lockdown for than once, and if a sentence can be provided with a date, it would be better. For instance, “Although in some countries (China, South Korea, Australia),  availability has been normalized for most categories of consumer products, except for medicines, various studies indicate that consumers still, after the lockdown period, feel concerned about  returning to stores [ 20, 21]”
2.    While writing hypotheses first explain the literature and then give the hypothesis. Giving all hypotheses in a paragraph is unorthodox way and not easy to read and understand. 
3.    If the author(s) first explain the literature and then provide the hypothesis, then the additional explaining (lines between 249-267) can be transferred to each relevant hypothesis.

Methodology:
1.    Please provide the response rate. Although 233 responses might be seen as a good response number, we don’t know the entire population. This rate is important to determine whether the responses can be generalized or not. 

Discussion:
The discussion session must be re-written because provided measures are generic suggestions. But this study conducted in 5 different countries. So the discussion session should answer the following questions:
1.    What are the country-specific measures?
2.    How each country can improve its SC in terms of sustainability? 
3.    What type of measurement can be taken for each stage of the supply chain (production, distribution, wholesale, and retail)?
4.    Also what is the theoretical contribution of this study?

Conclusion:
1.    In the conclusion part, it is stated that “some indicators important for food placement have been precisely isolated, with the definition of the intensity of their impact, observed overall at the level of the entire FSC, as well as by sectors.” What are those “some indicators”? Please explicitly write them. 

Language:
1.    Authors frequently used “etc” such as on line 28, 29, 40, 42, 52, 54, 66. Please remove some of them. 

Author Response

Dear Reviewer,

Thank you for your review. 

All of your suggestions and comments have been incorporated into the text. You can see the paragraphs in red colour in final version of article. Besides, here is you can find response to the reviewer.

Abstract:

  1. The originality of the study needs to be provided in the abstract section. – DONE. The originality of the study is indicated in the abstract. See the sentence in the abstract in red colour.

Introduction:

  1. Instead of mentioning some incentives which can be utilized to support the sustainability of the SC (on line 88-89), please explicitly state very important ones. – DONE. Very important incentives for sustainability of FCS have been stated. See the text in red colour in lines 93-96.
  2. Please write the originality of the study in the introduction. – DONE. The originality is clearly indicated in the Introduction. See the text in red colour in lines 79-84.

Theoretical backgrounds and hypothesis development:

  1. Some of the arguments for the first paragraph require date validation. Because due to pandemic, most of the countries has entered lockdown for than once, and if a sentence can be provided with a date, it would be better. For instance, “Although in some countries (China, South Korea, Australia), availability has been normalized for most categories of consumer products, except for medicines, various studies indicate that consumers still, after the lockdown period, feel concerned about  returning to stores [ 20, 21]” – DONE. The dates have been added where it was necassary to do so. See the marks in red in the first paragraph of the given chapter.
  2. While writing hypotheses first explain the literature and then give the hypothesis. Giving all hypotheses in a paragraph is unorthodox way and not easy to read and understand. – DONE. The chapter in question was restructured and hypotheses have been clearly derived from literature. See the paragraphs in red in lines 228-278.
  3. If the author(s) first explain the literature and then provide the hypothesis, then the additional explaining (lines between 249-267) can be transferred to each relevant hypothesis. – DONE. See paragraphs in red colour in lines 228-278.

Methodology:

  1. Please provide the response rate. Although 233 responses might be seen as a good response number, we don’t know the entire population. This rate is important to determine whether the responses can be generalized or not. – DONE. See the sentence in red colour in line 349.

Discussion:

The discussion session must be re-written because provided measures are generic suggestions. But this study conducted in 5 different countries. So the discussion session should answer the following questions:

  1. What are the country-specific measures?
  2. How each country can improve its SC in terms of sustainability?
  3. What type of measurement can be taken for each stage of the supply chain (production, distribution, wholesale, and retail)?
  4. Also what is the theoretical contribution of this study?

– DONE.  Discussion chapter was rewritten, see the text in red colour in lines 543-607. Measures listed were clearly defined and explained individually by all FSC sectors and analyzed WB countries. Theoretical contribution of the study has been explained.

Conclusion:

  1. In the conclusion part, it is stated that “some indicators important for food placement have been precisely isolated, with the definition of the intensity of their impact, observed overall at the level of the entire FSC, as well as by sectors.” What are those “some indicators”? Please explicitly write them. – DONE. Indicators have been clearly stated, see the text in red colour in lines 619-622.

Language:

  1. Authors frequently used “etc” such as on line 28, 29, 40, 42, 52, 54, 66. Please remove some of them. – DONE.

Thank you for your understanding and support.

With best regards

Authors

Reviewer 2 Report

Sustainability of Food Placement in Retailing during 3 COVID-19 Pandemic

The manuscript is interesting and well presented. The empirical study has correctly identified research gap in given geographical region and answered the research aim utilising hypothesis development and testing using FSC data. The study contributing by providing sustainability model of food placement in retail sector. 

The manuscript is well written and I recco. to accept it. 

Author Response

Dear Reviewer,

Thank you for your positive review. 

All of suggestions and comments from others reviewers have been incorporated into the text.

The complete text was proof-read.

Thank you for your understanding and support.

With best regard

Authors

Reviewer 3 Report

The methodology of empirical research should be described in more detail.

Conclusions should be expanded, pointing to the limitations of the analyzed problem and defining the directions of further research.

Author Response

Dear Reviewer,

Thank you for your review. 

All of your suggestions and comments have been incorporated into the text. You can see the paragraphs in red colour in final version of article. Besides, here is you can find response to the reviewer.

Comments and Suggestions for Authors:

English language and style (x) Moderate English changes required

– DONE. The complete text was proof-read.

The methodology of empirical research should be described in more detail. – DONE. Hypotheses have been clearly derived, the structure of sample was simplified and the questionnaire was done appropriately. See the text in red, especially lines 231-280; 316-317; 330-355.

Conclusions should be expanded, pointing to the limitations of the analyzed problem and defining the directions of further research – DONE. See the text in red in Conclusion, especially lines 628-631; 642-648; 652-654.

Thank you for your understanding and support.

With best regards

Authors

Reviewer 4 Report

I am not fully convinced with line 16-17. As the paper heading states it is about retail sector, however, results  consider all the supply chain elements. Probably needs to define food placement in the abstract itself.

Line 32 – delete “as a consequence”

Introduction section needs to be arranges in such a way that it addresses following questions: what is the issue, why is it an issue and how others have addressed the issue and how you will be addressing it and finally the structure of the paper.

Line 67 word focuses is in italics.

Line 71 word aims is in italics.

Line 72 abovementoned (separate it or add hypen)

Line 229-241 – Repetitive. Could just say that they all have significant impact.

Table 1 – What is the reason behind Production (managers) come from Croatia, Physical Distribution from Serbia only and so on?  I believe every country’s FSC comprises all the four elements of FSC. Is it the right way to do a research?

Table 2 – Headings to be in English. For example, Indikatori - Indicators

Author Response

Dear Reviewer,

Thank you for your review. 

All of your suggestions and comments have been incorporated into the text. You can see the paragraphs in red colour in final version of article. Besides, here is you can find response to the reviewer.

Comments and Suggestions for Authors:

I am not fully convinced with line 16-17. As the paper heading states it is about retail sector, however, results  consider all the supply chain elements. Probably needs to define food placement in the abstract itself. – DONE. See the explanation in the abstract, sentences marked in read colour (lines 15-16; 21-25). Since food placement and retail sector itself inseparable from other participants of food supply chain (FSC) the conducted study had to include all participants and FSC sectors in the region of Western Balkan.

Line 32 – delete “as a consequence” – DONE.

Introduction section needs to be arranges in such a way that it addresses following questions: what is the issue, why is it an issue and how others have addressed the issue and how you will be addressing it and finally the structure of the paper. – DONE. The introduction chapter has been restructured. Questions which are the paper's starting point are clearly indicated. See the text in red in Introduction, especially lines 31-36, 44-45, 77-87.

Line 67 word focuses is in italics. – DONE.

Line 71 word aims is in italics. – DONE.

Line 72 abovementoned (separate it or add hypen) – DONE.

Line 229-241 – Repetitive. Could just say that they all have significant impact. – DONE. Through a detailed view of literature, all research hypotheses regarding the impact of indicators on sustainability of food placement have been indicated precisely (lines 236-251) in line with Reviewer 1's requests.

Table 1 – What is the reason behind Production (managers) come from Croatia, Physical Distribution from Serbia only and so on?  I believe every country’s FSC comprises all the four elements of FSC. Is it the right way to do a research? – DONE.  The table in question was done appropriately so it may have lead to confusion. All FSC participants (sectors) have been analyzed within each WB country. See the new Table 1.

Table 2 – Headings to be in English. For example, Indikatori - Indicators. – DONE.

Thank you for your understanding and support.

With best regards

Authors

Round 2

Reviewer 4 Report

All my concerns have been addressed.